# A Pilot Study of the Effect of *Lactobacillus casei* Obtained from Long-Lived Elderly on Blood Biochemical, Oxidative, and Inflammatory Markers, and on Gut Microbiota in Young Volunteers

**DOI:** 10.3390/nu13113891

**Published:** 2021-10-29

**Authors:** Li-Hua Mei, Wen-Xuan Zheng, Zheng-Tao Zhao, Ning Meng, Qin-Ren Zhang, Wen-Jun Zhu, Rui-Ding Li, Xiao-Lin Liang, Quan-Yang Li

**Affiliations:** 1College of Light Industry and Food Engineering, Guangxi University, Nanning 530004, China; 1816301017@st.gxu.edu.cn (L.-H.M.); 2016301049@st.gxu.edu.cn (W.-X.Z.); zhengtaozhao85@gmail.com (Z.-T.Z.); 2016401006@st.gxu.edu.cn (N.M.); 2116401004@st.gxu.edu.cn (Q.-R.Z.); 1916393021@st.gxu.edu.cn (W.-J.Z.); 2016301014@st.gxu.edu.cn (R.-D.L.); 2National Engineering Laboratory for Cereal Fermentation Technology, Jiangnan University, 1800 Lihu Road, Wuxi 214122, China; 3Science Center for Future Foods, Jiangnan University, 1800 Lihu Road, Wuxi 214122, China

**Keywords:** probiotics, *Lactobacillus casei* LTL1879, healthy young people, blood biochemical, oxidant, inflammation, gut microbiota

## Abstract

Probiotic intake has been shown to improve certain physiological health indicators. We aimed to examine effects of *Lactobacillus* *casei* LTL1879, obtained from long-lived elderly volunteers, on blood biochemical, oxidative, and inflammatory markers and gut microbiota in twenty healthy, young volunteers. Volunteers were randomly divided into equal probiotic and placebo groups and changes in blood biochemical indicators, oxidative and inflammatory markers, and gut microbiota were examined after three weeks of probiotic intervention. The probiotic group’s antioxidant levels were significantly enhanced post-intervention. Total antioxidant capacity (T-AOC) levels were significantly increased (*p* < 0.0001), while malondialdehyde (MDA) levels decreased (*p* < 0.05), and total superoxide dismutase (T-SOD) levels increased, but with no significant difference. In addition, Interleukin-10 (IL-10) and tumor necrosis factor-α (TNF-α) levels were significantly up-regulated and down-regulated (*p* < 0.05, respectively). *Escherichia coli*, *Enterococcus*, and *Bacteroides* expression was significantly reduced (*p* < 0.05), while *Clostridium leptum*, *Bifidobacterium*, and *Lactobacillus* expression increased (*p* < 0.05). Volunteer health status was quantified using principal components and cluster analysis, indicating that the probiotic group’s overall score was higher than that of the placebo group. The results of this pilot study suggest *L. casei* LTL 1879 can significantly improve specific immune, oxidative, and gut microbiota characteristics related to health factors.

## 1. Introduction

It is believed that intestinal flora plays an important role in human health [1], and evidence shows its close relationship to human health and longevity. Recent research has illustrated that long-lived individuals have high intestinal microbial diversity and are rich in several potentially beneficial probiotic cultures, indicating a link between healthy aging and gut microbiota [2]. Moreover, it has been reported that the composition of the gut microbiota of healthy, long-lived individuals is significantly different from that of young individuals and the frail elderly [3,4,5,6]. Probiotic strains derived from long-lived individuals exhibited excellent antioxidant [7,8], anti-inflammatory, cholesterol-lowering [9], immune regulating [10], and anti-tumor activities [11].

Research regarding the adjustment of intestinal flora to improve human health has recently gained great interest. Previous studies have shown that by regulating the intestinal flora, the host’s capacity to resist pathogens can be improved, thereby strengthening the intestinal structure and immune system [12], and alleviating chronic inflammation [13]. Furthermore, intestinal flora shows evidence of plasma metabolism regulation [14] and healthy intestinal flora helps regulate cholesterol homeostasis and prevent or treat vascular disease [15]. Studies have proved that the intestinal flora controls type 2 immunity by inducing type T cells, balancing immune responses at the mucosal surface [16].

Intestinal flora can be regulated in various ways, including dietary intervention [17], multi-strain overall regulation, and individualized regulation of single strains. However, it is challenging to regulate the entire gut microbiome of the human body because of the complexity and heterogeneity of the human microbiome. Instead, partial regulation with screening strains may more feasible. It has been found that short-term probiotic supplementation enhanced the cellular immune function of healthy, elderly people [18], and supplementation with multiple strains of *Lactobacillus* probiotic improved some physiological indicators of patients with type 2 diabetes [19]. In addition, *Bifidobacterium bifidum* CCFM16 and *Lactobacillus plantarum* CCFM8610 have been reported to improve symptoms, flora, and immune response of patients with atopic dermatitis [20]. Recent research discovered that oral probiotics can play a role in the intestinal and systemic effects of COVID-19 [21]. Therefore, probiotics may be used as a potential therapy for regulating intestinal flora and improving health.

Although previous studies have shown the positive effects of probiotic intake on many health outcomes, many of these studies focused on patients and the elderly. Additionally, the health of young individuals is becoming a global concern because of increasing competition and pressure in the modern world. However, limited research has been conducted on the health effects of probiotic strains in young people, and the conclusions are not consistent [22,23]. In preliminary research, we successfully isolated *L. casei* LTL1879 from the feces of centenarians and demonstrated its potential probiotic properties in vitro. In the current study, we evaluated effects of the LTL1879 strain on the health of young individuals by examining blood biochemistry, oxidation, and inflammation markers; as well as monitoring gut microbiota changes in young volunteers before, during, and after continuous probiotic supplementation. Thereafter, a comprehensive score model was built to quantitatively analyze effects of probiotic strains on the health of young individuals.

## 2. Materials and Methods

### 2.1. LTL1879 Powder Preparation

*L. casei* LTL1879 strain was cultured on Man-Rogosa-Sharpe medium (MRS) agar plates and after single passage, the strain was inoculated in 2% MRS broth medium and incubated at 37 °C until the end of logarithmic growth (approximately 12 h), Cells were harvested by himac CR22N high-speed refrigerated centrifuge (Hitachi-Koki Co., Ltd., Ibaraki, Japan) and then weighed. Thereafter cells were mixed with the cryoprotectant (12% skimmed milk, 2% sucrose, and 2% trehalose) at a ratio of 1:10, and pre-frozen at −80 °C for 4 h. After pre-freezing, the mixture was thoroughly lyophilized using a vacuum freeze dryer (Shanghai Youpu Industrial Co., Ltd., Shanghai, China) (−50 °C and vacuumed < 15 Pa). The obtained probiotic powder was used for further experiments (the viable count of *L. casei* LTL1879 was 1.41 ± 0.12 × 10^11^ CFU/g). Cytoprotectants were lyophilized in the same way to serve as placebos for future experiments.

### 2.2. Participants and Study Design

A total of 20 healthy volunteers between the ages of 20–35 years were recruited from Nanning, China. All volunteers were in good health without gastrointestinal, metabolic, or digestive tract disease, bacterial or viral enteritis, or immunodeficiency. Volunteers did not consume products containing probiotics, antibiotics, or other drugs three months prior to the study. Volunteers’ basic information is presented in Table 1. The study was approved by the Institutional Review Board (or Ethics Committee) of Guangxi University (approval number GXU–2020–137), and all participants provided written informed consent.

Volunteers were randomly divided into a probiotic group (*n* = 10) and a placebo group (*n* = 10), and probiotic powder and placebo powder (Figure 1) were administered during the 3-week trial period. Volunteers were provided with a 2 g freeze-dried powder product daily, which needed to be taken with warm water on an empty stomach every morning. During the entire experimental period, volunteers were instructed to maintain their normal diet and work/rest habits without drinking or smoking. No products containing other probiotics or prebiotics were used. Anthropometric data and blood samples were collected before and after intervention. Fecal samples were collected at 0, 2, 3, and 4 weeks (one week post-intervention).

### 2.3. Anthropometric Assessment

Weight measurements were performed on an empty stomach. Volunteers were required to take off their shoes, hats, and coats, and take an upright posture, arms at the side, heels together, and eyes looking straight ahead. The height and body weight of volunteers were measured with a stadiometer (Shenzhen Mobil Electronics Co., Ltd., Shenzhen, China) to the nearest 0.1 cm and a calibrated scale (Shenzhen Mobil Electronics Co., Ltd., Shenzhen, China) to the nearest 0.1 kg respectively. Body mass index (BMI) was calculated as: BMI = weight (kg)/height (m^2^) [24]. A sphygmomanometer (Jiangsu Yuyue Medical Equipment Co., Ltd., Danyang, China) was used to measure blood pressure after resting for 5 min in the sitting position.

### 2.4. Blood Sample Collection and Biochemical Parameter Measurement

Fasting blood samples were collected by venipuncture extraction and stored in collection vessels with or without anticoagulants. Blood samples were left to stand for 30 min and then centrifuged at 1500× *g* at 4 °C for 15 min to obtain serum. Serum samples were stored at −80 °C for further analysis.

Biochemical parameters were analyzed at the Guangxi University Hospital. Fasting glucose, triglycerides (TG), high-density lipoprotein (HDL), low-density lipoprotein (LDL), creatinine, uric acid, alkaline phosphatase (ALP), aspartate transaminase (AST), and glutamyl transferase (GGT) levels were measured by Roche Cobas c702 automated biochemical analyzer (Roche Diagnostics, Basel, Switzerland).

### 2.5. Oxidative Marker Measurements

Malondialdehyde MDA, total superoxide dismutase (T-SOD), and total antioxidant capacity (T-AOC) serum levels were measured using assay kits (Nanjing Jiancheng Institute of Biotechnology, Nanjing, China) according to the manufacturer’s guidelines [25,26].

### 2.6. Inflammatory Marker Measurements

Interleukin-10 (IL-10) and tumor necrosis factor-α (TNF-α) serum levels were measured using enzyme-linked immunoassay kits (Shanghai Jianglai Biotechnology Co., Ltd., Shanghai, China) according to the manufacturer’s instructions.

### 2.7. Investigation of Gut Microbiota

#### 2.7.1. Fecal Sample Collection

Fresh feces were collected (≥5 g) the night before, or the morning of the visit. Fecal samples were stored in a sterilized 50 mL centrifuge tube at 4 °C, sent to the laboratory within 2 h after collection, and stored at −80 °C.

#### 2.7.2. Extraction of Fecal DNA

Bacterial DNA from fecal samples was extracted using a Stool Genomic DNA Extraction Kit (Solarbio, Beijing, China) following the manufacturer’s recommendations. Nucleic acids were obtained from 200 mg of fecal sample and eluted in 90 µL elution buffer solution. The concentration of DNA was measured using an Infinite M200 pro continuous wavelength multifunctional microporous detector (Tecan, Männedorf, Switzerland). Extracted DNA samples were stored at −80 °C.

#### 2.7.3. Real-Time PCR

Using real-time PCR, a total of seven bacteria were quantified from each fecal DNA sample. DNA Amplification and detection was performed using a Roche LIGHTCYCLER 96 real-time PCR instrument (Roche Diagnostics Co., Ltd., Basel, Switzerland) using optical-grade 96-well plates. Sample analysis was routinely performed in a total volume of 20 μL using SYBR Green qPCR Master Mix (Vazyme Biotech Co. Ltd., Nanjing, China). Each reaction included 2 μL of template DNA, 7 μL ddH_2_O,10.0 μL of 2 × ChamQ Universal SYBR qPCR Master Mix (Vazyme Biotech Co., Ltd., Nanjing, China), 0.5 μL of primer 1 and primer 2 with a concentration of 10 μM, Real-time PCR conditions consisted of an initial denaturation step at 95 °C for 5 min and an amplification step, followed by 40 cycles of denaturation at 95 °C for 30 s, annealing at optimum annealing temperature of primers (Table 2) for 30 s, elongation at 72 °C for 1 min, and re-elongation at 72 °C for 8 min. At the end of the PCR assay, a dissociation curve analysis was performed to check for non-specific products and/or SYBR Green probe contamination. Relative quantification was used, and the relative expression of the strain was used as the standard. The formula is as follows:Relative expression level = 2 ^−{(Ct value of target gene to be tested − Ct value of internal reference gene to be tested) − (Ct value of control target gene − Ct value of control internal reference gene)}^(1)

### 2.8. Statistical Analysis of Comprehensive Health Indicators and Quantitative Evaluation of Health Indicators

Principal component analysis (PCA) was used to quantitatively analyze volunteer health data, extract data features, and reduce dimensionality from the three dimensions of antioxidant indicators, inflammation indicators, and intestinal flora to obtain a comprehensive health evaluation index, The health status of volunteers was reflected by the quantitative expression of the comprehensive health evaluation index.

### 2.9. Statistical Analysis

Statistical analyses were performed using SPSS (version 26.0; SPSS, Inc., Chicago, IL, USA) and GraphPad Prism 7 (GraphPad Software, San Diego, CA, USA). Changes (mean SD) in individual health indices and microbiota between the baseline, during, and post intervention periods were assessed with a paired *t*-test. Differences were considered statistically significant at *p* < 0.05. Spearman rank correlation was used to determine the correlation between *Lactobacillus*, antioxidation, and inflammation levels.

## 3. Results

### 3.1. Effects of LTL1879 Intervention on Blood Parameters and Anthropometric Measurements

Blood parameters and anthropometric measurements are summarized in Table 3. Body weight, BMI, and blood pressure did not differ significantly between the baseline and after 3 weeks of intervention measurements. Fasting blood glucose (FBG), low-density lipoprotein (LDL), uric acid (UA), creatinine (Cre), alkaline phosphatase (ALP), aspartate aminotransferase (AST), and glutamyl transferase (GGT) levels decreased after 3 weeks of intervention compared to baseline values, but changes were not statistically significant (*p* > 0.05). The results showed small fluctuations in the volunteers’ blood biochemical parameters after 3 weeks of intervention. However, all parameters were within normal range and their values were not statistically significant.

### 3.2. Effect of LTL1879 Intervention on Oxidation Indexes

Figure 2 summarizes serum oxidation indices of volunteers. The probiotic group and the placebo group had the same serum SOD activity level (137.91 ± 6.56 U/mL vs. 137.28 ± 11.3 U/mL) (Figure 2A) at baseline. After 3 weeks of intervention, the serum SOD activity of the probiotic group increased slightly, while that of the placebo group decreased, but differences were not significant (*p* > 0.05) (Figure 2A). MDA levels of the probiotic group significantly decreased by 19.28% (*p* < 0.05) after the intervention (Figure 2B). In contrast, no significant changes in MDA levels in the placebo group (3.09 ± 0.29 vs. 3.11 ± 0.29, *p* > 0.05) were observed. Similarly, the serum T-AOC activity of the probiotic group was significantly increased by 52.70% (*p* < 0.0001) (Figure 2C), while T-AOC activity in the placebo group did not change significantly (11.66 ± 1.73 vs. 11.29 ± 1.49%, *p* > 0.05).

Sex-based analysis of volunteers in the probiotic group indicated that there was no difference in serum T-SOD, MDA, and T-AOC activity between male and female volunteers at baseline. After three weeks of intervention, the MDA levels in men decreased by 25.00% (*p* < 0.01), which was significantly higher than that in women (13.50%) (*p* < 0.05) (Figure 2b). No sex difference was observed in T-SOD and T-AOC changes after intervention (Figure 2c).

### 3.3. Effect of LTL1879 Intervention on Immune Parameters

To evaluate the effect of LTL1879 on inflammation markers in volunteers, levels of pro-inflammatory marker TNF-α and anti-inflammatory marker IL-10 were examined (Figure 3). After 3 weeks of probiotic intervention, serum IL-10 levels in the probiotic group significantly increased by 8.10% (*p* < 0.05), while no changes were observed for the placebo group (Figure 3A). The TNF-α level decreased by 32.38% (*p* < 0.05) and 14.49% (*p* < 0.01) in the probiotic and placebo group, respectively. These results demonstrated that the inclusion of probiotics may improve the immunity of young, healthy individuals.

The sex-based differences in inflammatory markers in the probiotic group were also investigated, as shown in Figure 3a,b. In general, females exhibited higher IL-10 levels than males and no significant changes were observed after intervention. In contrast, TNF-α levels decreased for both male and female volunteers, however, no difference between male and female TNF-α levels were observed.

### 3.4. Effect of LTL1879 Intervention on Typical Fecal Microorganisms

In this study, the total intestinal flora was used as the internal reference gene, and *E. coli*, *Bacteroides*, *Clostridium leptum*, *Enterococcus*, *Bifidobacterium*, and *Lactobacillus* were the target bacterial genera. The relative fecal expression of the six important bacterial groups in volunteers during and after LTL1879 intervention is shown in Figure 4. In the placebo group, *E. coli* increased by 9.79% in the fourth week of intervention (*p* < 0.05), while *Lactobacillus* increased by 16.83% in the third week of the intervention (*p* < 0.05) (Figure 4A). No changes were detected in the other bacterial groups (*p* > 0.05). In contrast, in the probiotic group, expression of all six bacterial genera significantly changed. The expression levels of *E. coli*, *Bacteroides*, and *Enterococcus* began to decline after the second week of intervention, and were further reduced by 33.54% (*p* < 0.0001), 29.75% (*p* < 0.01), and 33.66% (*p* < 0.001) after 3 weeks of intervention (Figure 4A). These three bacterial groups rebounded after probiotic intervention was stopped for one week, but the relative expression of *Bacteroides* was still 22.40% lower than the baseline value (*p* < 0.01) (Figure 4A). The expression levels of *Clostridium leptum*, *Bifidobacterium*, and *Lactobacillu**s* were significantly increased by 30.97% (*p* < 0.05), 27.32% (*p* < 0.05), and 44.27% (*p* < 0.001) in the second week of intervention (Figure 4A), These changes were more notable during the third week of probiotic intervention, where *Clostridium leptum*, *Bifidobacterium*, and *Lactobacillus* expression were significantly increased by 67.92% (*p* < 0.05), 74.09% (*p* < 0.01), and 107.20% (*p* < 0.001), respectively (Figure 4A). After probiotic intervention was stopped for one week, the relative expression of *Clostridium leptum*, *Bifidobacterium*, and *Lactobacillus* was still higher than that of the baseline value. For instance, the expression of *Lactobacillus* was significantly higher than the baseline value by 42.45% (*p* < 0.05) (Figure 4A) after stopping the intervention for 1 week.

The sex-based differences in LTL1879 influence on intestinal microbes were investigated. No difference in expression levels of the six bacterial genera (*p* > 0.05) between male and female volunteers were observed before probiotic intervention (Figure 4B). During the second week of intervention, significant differences in male and female expression of *Bacteroides*, *Bifidobacterium*, and *Lactobacillus* were detected (*p* < 0.05) (Figure 4B). The expression level of *Bacteroides* in females reduced by 18.18% (*p* < 0.05) compared to 5.88% increase in males (*p* > 0.05) (Figure 4B). Similarly, *Bifidobacterium* expression in females increased by 47.52% (*p* < 0.05), while expression in males increased by 6.93%. The expression of *Lactobacillus* significantly increased by 52.04% (*p* < 0.05) and 38.23% (*p* < 0.05) in females and males, respectively (Figure 4B).

### 3.5. Correlation between Lactobacillus, Oxidative, and Inflammatory Markers

After 3 weeks of LTL1879 intervention, changes in serum oxidative and inflammatory markers, and intestinal microbes significantly improved. Therefore, a correlation analysis was performed between the expression of *Lactobacillus,* serum oxidative, and inflammatory markers in the probiotic group. As shown in Figure 5A,B, *Lactobacillus* expression was negatively correlated with pro-inflammatory marker TNF-α levels (*p* < 0.01), while it was positively correlated with anti-inflammatory marker IL-10 level, but the correlations were not significant (*p* > 0.05). Significant positive correlations were found between *Lactobacillus* expression and T-SOD and T-AOC activity (*p* < 0.01 and *p* < 0.0001, respectively) (Figure 5C,E). In addition, a significant negative correlation between *Lactobacillus* expression and MDA levels (*p* < 0.001) was observed (Figure 5D).

### 3.6. Quantitative Evaluation of Volunteer Health Status

Using PCA and dimensionality reduction, multiple health indicators of volunteers were measured using comprehensive indicators. PCA was performed on 11 health indicators of 20 volunteers, and the results are shown in Table 4. Three principal components were extracted after analysis, which had characteristic values > 1, and their cumulative variance contribution rate reached 71.468%. Therefore, we replaced the original 11 indicators with the three extracted principal components to evaluate volunteer health status.

Table 5 shows the principal component load matrices of the 11 health indicators of the volunteers. The matrix reflects the relative magnitude and direction of the health indicators’ main component load. The load of the first principal component was relatively high, among which T-AOC and *Lactobacillus* had load values of 0.940 and 0.879, respectively. These two indices have a positive effect on the first principal component. In the second principal component, IL-10 had the largest load value (0.790). *Bifidobacterium* and *Enterococcus* had a negative impact on the second principal component. In the third principal component, T-SOD was the only factor that exhibited a positive effect.

To eliminate the influence of the dimensional difference of different index data, the raw data of each health index were standardized and transformed into dimensionless data with a mean value of 0 and a standard deviation of 1. The score of each principal component was calculated according to the standardized indicators and factor loading matrix using the following formula:F_1_ = 0.236X_1_ + 0.353X_2_ + 0.397X_3_ + 0.113X_4_ + 0.158X_5_ + 0.318X_6_ + 0.283X_7_ + 0.289X_8_ + 0.339X_9_ + 0.327X_10_ + 0.371X_11_(2)
F_2_ = 0.084X_1_ + 0.037X_2_ + 0.077X_3_ + 0.714X_4_ + 0.327X_5_ + 0.304X_6_ − 0.116X_7_ − 0.178X_8_ − 0.296X_9_ − 0.378X_10_ + 0.043X_11_(3)
F_3_ = 0.502X_1_ − 0.187X_2_ − 0.042X_3_ − 0.420X_4_ + 0.710X_5_ − 0.081X_6_ − 0.040X_7_ + 0.045X_8_ − 0.084X_9_ − 0.107X_10_ − 0.035X_11_(4)

Taking the three principal components and the proportion of the feature value corresponding to each principal component, to the sum of the total feature values of all extracted principal components as weights, the principal component comprehensive model is calculated as follows:F_sum_ = 0.509F_1_ + 0.111F_2_ + 0.095F_3_(5)

Based on the PCA, the comprehensive health index F_sum_ of different volunteers was obtained according to the above-mentioned comprehensive score model. The results are listed in Table 6. The comprehensive health index was positively correlated with the health status of the volunteer. Among the 20 volunteers, volunteers with serial numbers 1, 2, 5, 7, 8, 9, 12, 15, 17, and 19 belonged to the probiotic group, and volunteers with serial numbers 3, 4, 6, 10, 11, 13, 14, 16, 18, and 20 were in the placebo group. According to the ranking results, volunteers from the probiotic group had a higher comprehensive health index than the placebo group. This result shows that LTL1879 can improve the health of volunteers.

## 4. Discussion

Among the gut bacteria that act as functional organisms in humans, probiotics are worth studying because they have a wide range of beneficial effects on host health and longevity [34]. Probiotics obtained from long-lived elderly individuals have attracted much attention in recent years because of their excellent probiotic effects. However, reports mostly focused on evaluating the effects of probiotics on specific groups such as patients and the elderly, and few studies have proven the effects of probiotics on healthy, young individuals. In this study, short-term probiotic intervention was conducted on healthy, young volunteers, and it was found that *L. casei* LTL1879 obtained from long-lived elderly individuals improved specific oxidative, immune, and intestinal microbial characteristics related to volunteer health.

In this study, no significant changes were found in blood parameters and anthropometric measurements of the volunteers after 3 weeks of probiotic intervention. This may be related to intervention duration, as well as volunteer age and baseline health. Normally, blood parameters and anthropometric measurements of healthy individuals are relatively stable. Studies have shown that the probiotic effect on volunteers’ blood parameters and anthropometric measurements are related to baseline health status [35,36]. When the volunteers’ baseline health status was poor, long-term probiotic intervention may provide significant improvement [37,38,39]. The volunteers in this study were healthy, and all indicators were within the normal range prior to intervention. The short-term intervention of LTL1879 may not be sufficient to cause changes in blood parameters and anthropometric measurements.

The imbalance between antioxidants and pro-oxidants is called oxidative stress and is related to various non-communicable diseases [40]. Probiotic supplementation has been reported to reduce oxidative stress and improve antioxidant indices [41,42]. In this study, after 3 weeks of LTL1879 intake, the serum SOD level of volunteers increased, but no significant results were obtained. A study in healthy volunteers showed that probiotic *Lactobacillus* increased serum SOD levels after 4 weeks of intervention significantly [43]. In contrast, Valentini et al. [44] discovered that supplementing healthy elderly volunteers with a variety of probiotics for 8 weeks did not affect the activity of SOD. The different results between these studies may be related to variances in probiotic strains, doses, or differences between volunteers. In addition to endothelial damage, increased oxidative stress increases lipid oxidation and leads to an increase in MDA in the body. Therefore, MDA produced by lipid peroxidation can reflect the level of oxidative stress in the body [45]. In this study, MDA levels in the probiotic group was significantly reduced, which is in agreement with results from previous studies [36,46,47,48,49]. Studies have shown that probiotics can change the lipid profile and the reduction in MDA may be due to an improvement in the lipid profile [50]. The main constituents of the T-AOC structure are vitamin E, SOD, and glutathione [51]. However, the significant increase in T-AOC in this study did not seem to be related to SOD. T-AOC is also affected by various nutritional, genetic, and environmental factors; therefore, the possible mechanism underlying the increase in T-AOC levels remains to be explored in further studies. In summary, the beneficial changes in oxidative stress parameters are only reflected in the probiotics group, which indicates that LTL1879 improved specific oxidative characteristics related to health factors significantly.

Increased levels of inflammatory markers have been proved associated with frailty and mortality [52]. Reducing low-grade inflammation may be a way to reduce or prevent the onset and severity of some diseases. It has been reported that probiotics induce pro-inflammatory factor TNF-α and induce anti-inflammatory factor IL-10 in human monocytes in a dose-dependent manner [53]. Similar results were observed in the present study. After 3 weeks of LTL1879 supplementation, volunteer IL-10 levels significantly increased, while TNF-α decreased. As a functional result of the interaction between commensal microorganisms, parenchymal, and immune cells at the mucosal interface [54], IL-10 exerts immunosuppressive or immunostimulatory effects on many types of cells. TNF-α encodes tumor necrosis factors produced by monocytes and macrophages. Increased levels of TNF-α usually aggravate the degree of inflammation in the body. Mousavi et al. [55] confirmed in a meta-analysis that the intake of probiotics reduced TNF-α levels in healthy volunteers. Therefore, LTL1879 can improve specific immune characteristics related to health factors significantly.

The expression levels of six important bacterial genera in the feces of volunteers were selected and followed [56,57]. We found that after 3 weeks of LTL1879 supplementation, the expression levels of *Lactobacillus*, *Clostridium leptum*, and *Bifidobacterium* were significantly and continuously increased. In contrast, expression levels of *E. coli*, *Bacteroides*, and *Enterococcus* continuously decreased. One week after probiotic supplementation was ceased, the expression levels of these six bacterial groups all tended to return to baseline level, but their expression was still increased compared to the baseline level. During probiotic intervention, the expression of *Lactobacillus* notably increased. Studies have shown that an increased amount of intestinal *Lactobacillus spp.* is an important defense factor against intestinal infections [58,59], and the increase in the abundance of *Lactobacillus* resulting from probiotic supplementation maintained or improved intestinal function and reduced inflammation in aging individuals [60]. Another study showed that the administration of *L. casei* Shirota is associated with a significant increase in *Bifidobacterium* expression in healthy, middle-aged men [61], which indicated a beneficial effect on intestinal flora stability [62]. Moreover, the reduction of certain bacteria, such as *E. coli*, may protect the integrity of the intestinal barrier and promote intestinal health. Supplemented probiotics are part of the “transient microbiota” in our body for a relatively short period, and their permanent colonization is largely hindered by the resident flora [63]. The results of this study were similar to those of previous studies [64]; after probiotic supplementation was stopped, the genus recovered but did not immediately return to baseline level, indicating that LTL1879 could be colonized in volunteers to a certain extent and continue to play a role. In addition, this study found that after 2 weeks of LTL1879 intake, there were significant differences in the expression of *Bacteroides*, *Bifidobacterium*, and *Lactobacillus* in males and females, where the expression in females was significantly increased than that in men. It has been reported that sex has an important influence on the classification and combination of gut microbes in animal models [65,66] and may be caused by sex hormone level differences [67,68].

After 3 weeks of LTL1879 supplementation, the change in intestinal *Lactobacillus* expression was noted clearly. Since LTL1879 is a *Lactobacillus* strain and the change in *Lactobacillus* expression was not reflected in the placebo group, we hypothesize that the change in *Lactobacillus* expression in the probiotic group is the result of the direct effect of LTL1879. This study found that a correlation between the expression of *Lactobacillus* and oxidative and inflammatory markers. Studies have shown that probiotic *Lactobacillus* reduced intestinal dysfunction and inflammation caused by TNF-α [69]. In this study, the increase in *Lactobacillus* expression was significantly negatively correlated with TNF-α, which is consistent with the results of Chen et al. [60], who found that the increase in *Lactobacillus* abundance due to the addition of *Lactobacillus paracasei* was negatively correlated with TNF-α. The results showed that the improvement of immune and oxidative indices of volunteers may be related to intestinal microorganism changes caused by LTL1879, but the possible mechanism remains to be explored in further studies.

To objectively and comprehensively, evaluate the impact of LTL1879 on healthy young volunteers, we examined the comprehensive impact of multiple variables on volunteer health status. We normalized the health-related immune, oxidation, and gut microbial indicators and PCA was used to explore and quantify the effects of probiotics on health. To filter the superimposed influence of the synergy between different indicators on the final quantitative results, linear changes were used to simplify multiple variables into a few comprehensive variables for evaluation. The comprehensive health index may reflect the impact of probiotics on the health of each volunteer to a certain extent. Based on the indicators measured in this study, it was found that the health index of volunteers in the probiotic group was better than that in the placebo group. This phenomenon was common in every individual.

The effects of probiotic strains, prebiotic compounds, and/or their combinations are different, and depend on the expected duration of the intervention, subject population, and its targeting mechanism. The results of this pilot study suggest *L. casei* LTL1879 can significantly improve specific immune, oxidative, and gut microbiota characteristics related to health factors. However, the number of participants in this study is relatively small, and the effect of exploring strains on individuals was limited. Therefore, it is necessary to expand the intervention population, select targeted models, and extend the intervention time in future studies, to further determine the function of LTL1879.

## 5. Conclusions

A 3-week pilot intervention with *L. casei* LTL1879 improved specific immune, oxidation, and intestinal microbial indicators of healthy, young volunteers. Compared to the placebo group, LTL1879 showed a positive effect on the serum oxidative markers (T-SOD, MDA, and T-AOC) and inflammatory markers (IL-10 and TNF-α). We observed that even short-term LTL1879 intake can affect gut microbes of volunteers. However, anthropometric measurements and blood indicators of volunteers were not significantly affected. LTL1879 supplementation merits further research as a potential strategy for maintaining individual health.

## Figures and Tables

**Figure 1 nutrients-13-03891-f001:**
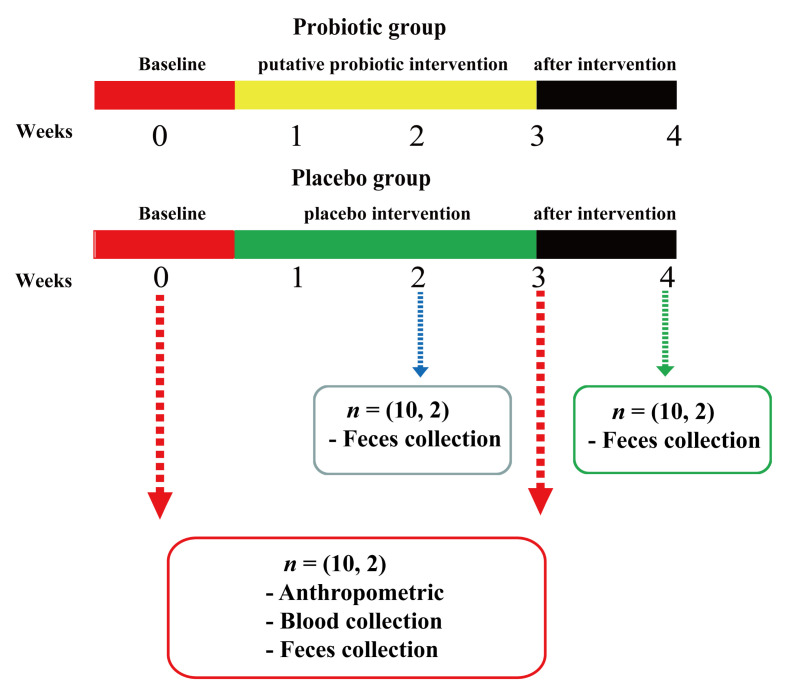
Schematic diagram indicating probiotic and placebo group intervention of the volunteers. Each group took the probiotic or placebo for three weeks. Blood samples of volunteers were collected at 0 and 3 weeks, and fecal samples were collected at 0, 2, 3, and 4 weeks.

**Figure 2 nutrients-13-03891-f002:**
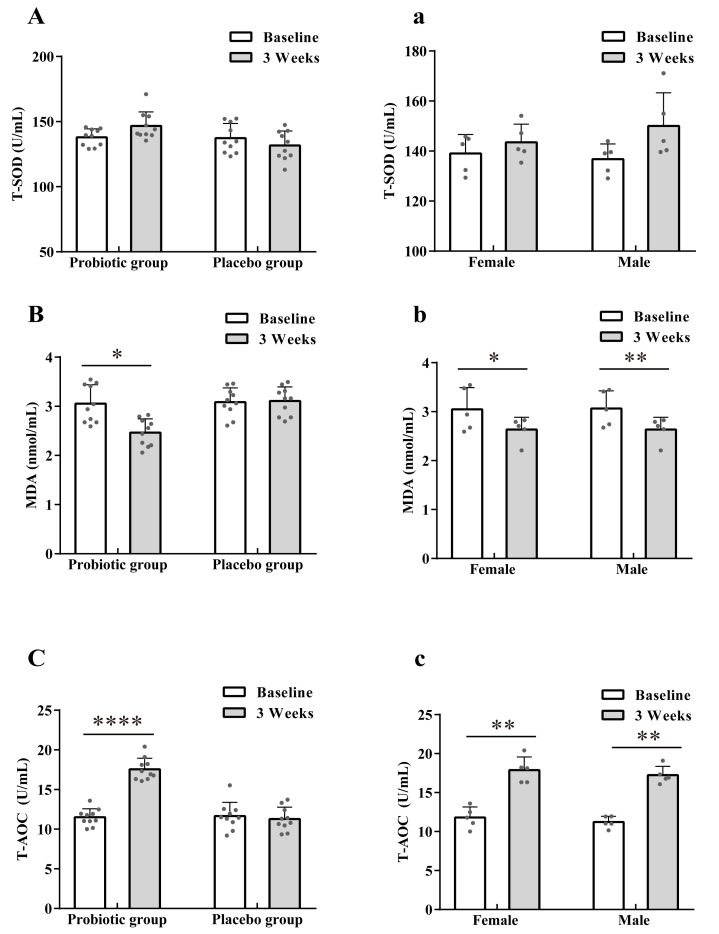
Effects of LTL1879 on total superoxide dismutase (T-SOD) activity, malondialdehyde (MDA) levels, and total antioxidant capacity (T-AOC) activity in serum of volunteers. (**A**) Serum T-SOD activity of volunteers. (**B**) Serum MDA levels of volunteers. (**C**) Serum T-AOC activity of volunteers. (**a**) Serum T-SOD activity of volunteers in probiotics group. (**b**) Serum MDA levels of volunteers in probiotics group. (**c**) Serum T-AOC activity of volunteers in the probiotics group. Data are shown as means standard deviations (SD). Asterisks indicate significant differences (paired *t*-test, * *p* < 0.05, ** *p* < 0.01, **** *p* < 0.0001). *n* = 10 or *n* = 5 volunteers per group.

**Figure 3 nutrients-13-03891-f003:**
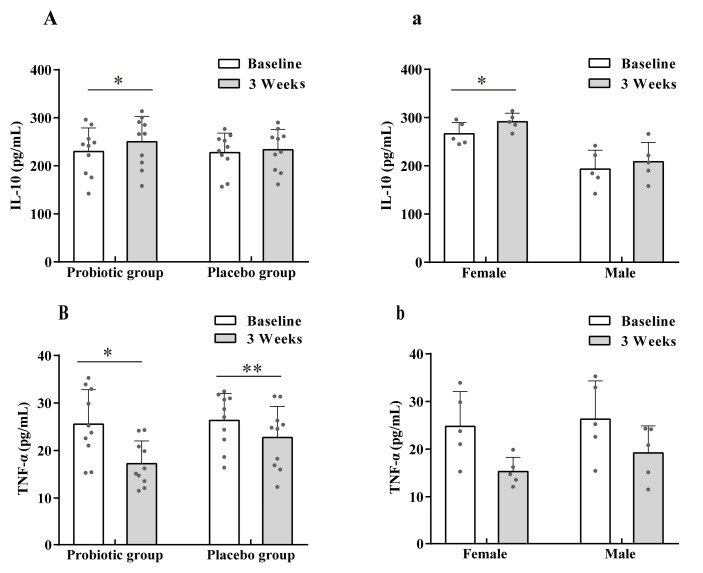
Effect of LTL1879 on inflammatory factors interleukin-10 (IL-10) and tumor necrosis factor-α (TNF-α). (**A**) Serum IL-10 levels in volunteers. (**B**) Serum TNF-α levels in volunteers. (**a**) Serum IL-10 levels in the probiotics group. (**b**) Serum TNF-α levels in the probiotics group volunteers. Data are shown as means standard deviations (SD). Asterisks indicate significant differences (paired *t*-test, * *p* < 0.05, ** *p* < 0.01). *n* = 10 or *n* = 5 volunteers per group.

**Figure 4 nutrients-13-03891-f004:**
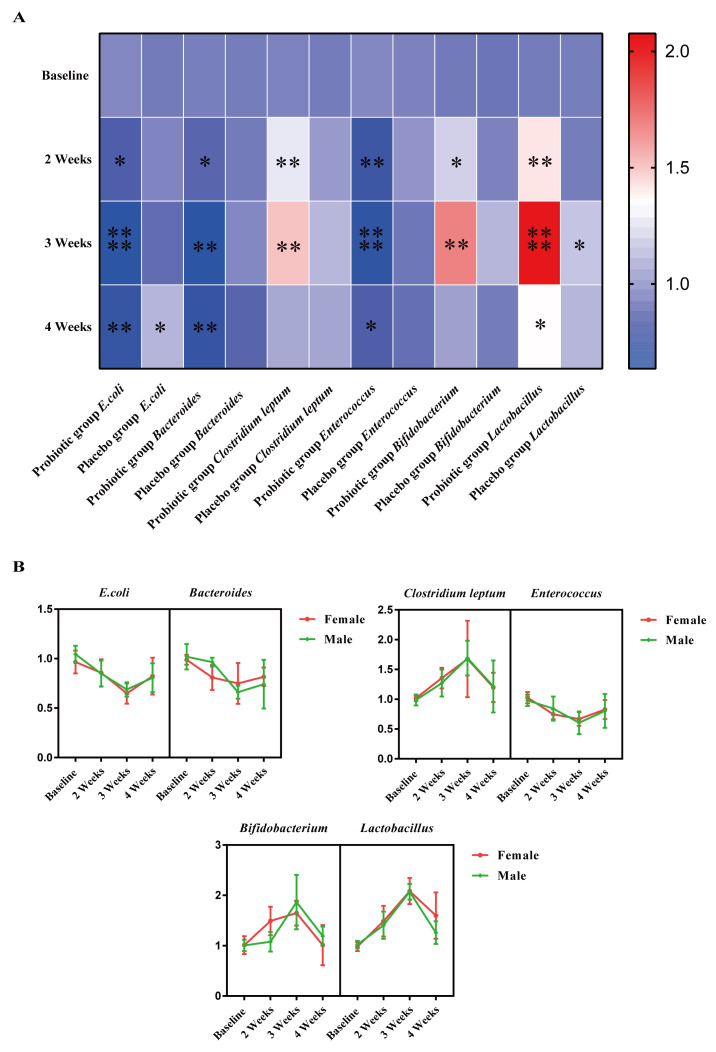
Relative expression of fecal flora in volunteers. (**A**) Relative expression levels of 6 species of bacteria in the feces of volunteers (**B**) Relative expression levels of 6 species of bacteria in the feces of volunteers in the probiotics group. Colors ranging from blue to red indicate low to high expression. Asterisks indicate significant differences (paired *t*-test, compared with baseline value * *p* < 0.05, ** *p* < 0.01, **** *p* < 0.0001). *n* = 10 or *n* = 5 volunteers per group.

**Figure 5 nutrients-13-03891-f005:**
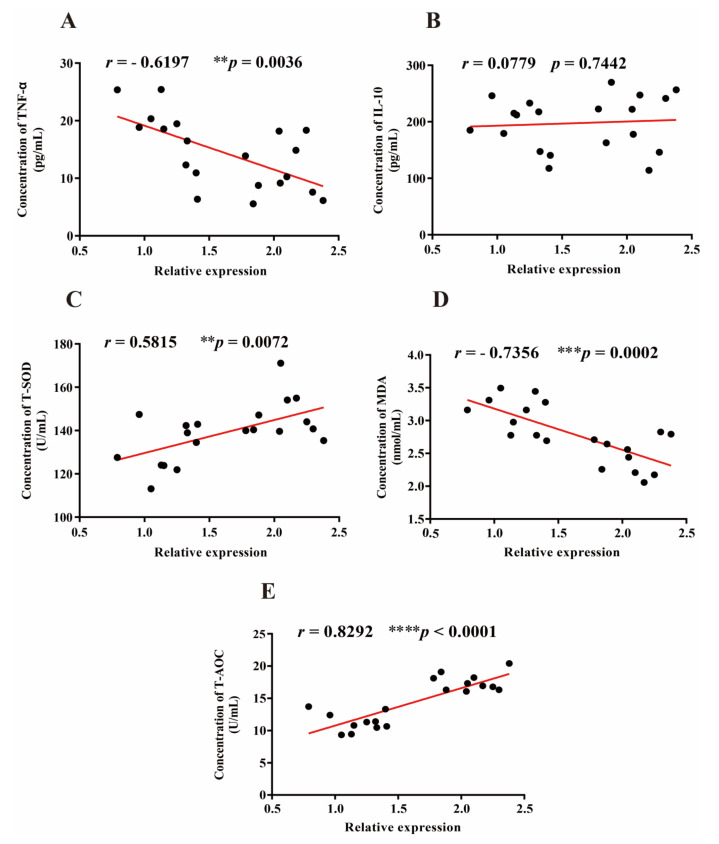
Correlation between the expression of *Lactobacillus*, oxidative, and inflammatory markers. Rank tests with Spearman’s correlation coefficient were used to assess correlations between *Lactobacillus* and oxidative markers (T-SOD, MDA, T-AOC), pro-inflammatory marker (TNF-α), and anti-inflammatory marker (IL-10). Asterisks indicate significant differences (Compared with baseline values, ** *p* < 0.01, *** *p* < 0.001, **** *p* < 0.0001). *n* = 10 volunteers per group.

**Table 1 nutrients-13-03891-t001:** Basic characteristics at baseline of the 20 subjects involved in the study.

	Probiotic Group	Placebo Group	*p*-Value
Age	24.9 ± 1.66	23.9 ± 0.99	0.120
Height (m)	1.67 ± 0.07	1.66 ± 0.09	0.828
Weight (kg)	59.36 ± 7.4	60.77 ± 19.53	0.835
Male/Female	5/5	4/6	-

All values are presented as mean (SD).

**Table 2 nutrients-13-03891-t002:** Primer sequences of 7 species.

Bacteria	Primer Sequence (5′–3′)	Annealing Temperature	References
Total intestinal flora	F: ACTCCTACGGGAGGCAGCAG R: ATTACCGCGGCTGCTGG-3′	64 °C	Cheng, Y. et al. [27]
*Escherichia coli*	F: GTTAATACCTTTGCTCATTGA R: ACCAGGGTATVTTAATCCTGTT	60 °C	Nelson, E. A. et al. [28]
*Bifidobacterium*	F: GGGTGGTAATGCCGGATG R: CCACCGTTACACCGGGAA	65 °C	Liu, R. [29]
*Lactobacillus*	F: AGCAGTAGGGAATCTTCCA R: ATTTCACCGCTACACATG	62 °C	Walter, J et al. [30]
*Enterococcus*	F: CCCTTATTGTTAGTTGCCATCATT R: ACTCGTTGTTGTACTTCCCATTGTT	60 °C	Rinttilä, T. et al. [31] Nelson, E. A. et al. [28]
*Bacteroides*	F: CTGAACCAGCCAAGTAGCG R: CCGCAAACTTTCACAACTGACTTA	68 °C	Pang, X. et al. [32]
*Clostridium leptum*	F: CCCTTCAGTGCCGCAGT R: GTCGCAGGATGTCAAGAC	58 °C	Wu, X. et al. [33]

**Table 3 nutrients-13-03891-t003:** Changes of blood parameters and anthropometric measurements in two groups under LTL1879 intervention.

Parameter	Probiotic Group			Placebo Group		
	Baseline	3 Weeks	*p-*Value	Baseline	3 Weeks	*p-*Value
Weight (kg)	59.36 ± 7.4	59.72 ± 7.41	0.300	60.77 ± 19.53	61.19 ± 19.19	0.629
BMI (kg/m^2^)	21.25 ± 2.12	21.36 ± 1.97	0.360	21.64 ± 5.13	21.80 ± 5.00	0.569
Systolic blood pressure (mmHg)	107.00 ± 11.54	109.90 ± 8.67	0.483	110.40 ± 17.79	112.80 ± 12.95	0.227
Diastolic blood pressure (mmHg)	68.40 ± 6.57	70.90 ± 7.80	0.426	73.50 ± 11.70	75.30 ± 7.27	0.466
FBG (mmol/L)	4.86 ± 0.4	4.73 ± 0.43	0.340	4.87 ± 0.33	4.9 ± 0.52	0.855
Triglyceride (mmol/L)	1.04 ± 0.17	1.05 ± 0.28	0.862	1.5 ± 1.61	1.99 ± 2.26	0.058
HDL (mmol/L)	1.42 ± 0.36	1.7 ± 0.38	0.078	1.47 ± 0.36	1.63 ± 0.46	0.455
LDL (mmol/L)	2.43 ± 0.33	2.42 ± 0.53	0.952	2.98 ± 0.94	2.75 ± 0.88	0.115
UA (μmol/L)	382.4 ± 56.87	342.6 ± 111.03	0.313	426.9 ± 135.51	428.6 ± 74.69	0.939
Creatinine (μmol/L)	79.8 ± 11.34	78.1 ± 22.06	0.746	82.1 ± 17.89	78.4 ± 18.49	0.387
ALP (U/L)	73.2 ± 17.33	70.4 ± 19.65	0.612	72.8 ± 18.12	72.9 ± 13.27	0.984
AST (U/L)	21.1 ± 7.5	17.8 ± 6.25	0.133	24.4 ± 9.03	22.6 ± 5.52	0.420
GGT (U/L)	18.4 ± 8.58	11.2 ± 3.43	0.052	23.7 ± 16.68	27.6 ± 13.72	0.200

All values are presented as mean (SD). Abbreviations: BMI, body mass index; FBG, fasting blood glucose; HDL, high-density lipoprotein; LDL, low-density lipoprotein; UA, Uric acid; ALP, alkaline phosphatase; AST, Aspartic acid transaminase; GGT, glutamyl transferase.

**Table 4 nutrients-13-03891-t004:** Eigenvalues and cumulative variance contribution rates of health status assessment fac-tors in volunteers.

Principal Component	Eigenvalues	Variance Contribution/%	Accumulative Variance Contribution/%
1	5.596	50.870	50.870
2	1.226	11.145	62.015
3	1.040	9.453	71.468

**Table 5 nutrients-13-03891-t005:** The factor load matrix of the principal component on each health index.

Health Index	Principal Component		
	1	2	3
T-SOD	0.558	0.093	0.512
MDA	0.834	0.041	−0.191
T-AOC	0.940	0.085	−0.043
IL-10	0.267	0.790	−0.428
TNF-α	0.373	0.362	0.724
*E. coli*	0.752	0.337	−0.083
*Bacteroides*	0.670	−0.128	−0.041
*Clostridium leptum*	0.684	−0.197	0.046
*Enterococcus*	0.802	−0.328	−0.085
*Bifidobacterium*	0.773	−0.418	−0.109
*Lactobacillus*	0.879	0.047	−0.036

**Table 6 nutrients-13-03891-t006:** Health status prediction and evaluation results of volunteers.

Number	F_1_	F_2_	F_3_	F_sum_	Rank
1	2.396	0.178	0.166	1.255	1
2	1.949	0.579	0.568	1.110	7
3	0.575	0.177	0.146	0.326	19
4	0.926	0.058	0.172	0.494	13
5	2.448	−0.035	−0.205	1.223	5
6	1.110	0.622	0.292	0.662	11
7	1.807	0.176	−0.385	0.903	10
8	2.173	0.533	0.274	1.191	6
9	2.286	0.875	−0.149	1.247	2
10	0.800	0.395	0.074	0.458	17
11	1.040	0.548	0.181	0.607	12
12	2.267	0.227	0.649	1.241	3
13	0.756	0.683	0.263	0.485	14
14	0.765	0.455	0.260	0.465	16
15	2.252	0.257	0.660	1.237	4
16	0.758	−0.041	0.304	0.410	18
17	1.921	0.517	0.044	1.039	8
18	0.524	0.316	0.152	0.316	20
19	1.791	0.780	0.428	1.039	9
20	0.783	0.510	0.281	0.482	15

## Data Availability

The data in this study are available on request from the author.

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
