# Peer review of "A Pilot Study of the Effect of Lactobacillus casei Obtained from Long-Lived Elderly on Blood Biochemical, Oxidative, and Inflammatory Markers, and on Gut Microbiota in Young Volunteers"

_nutrients, 2021, doi:10.3390/nu13113891_

Round 1

Reviewer 1 Report

Dear Sirs,

this is a really interesting studies with intriguing results. However, English language needs to be significantly improved. In addition, could the authors possibly inform us about any existing differences or not in those people having received this probiotic in the abundance of Akkermansia muciniphila and Faecalibacterium prausnitzii, which are well known for their beneficial effects on human's health? 

Reviewer 2 Report

This manuscript describes an interesting study of the effect of Lactobacillus casei strain derived from a centenarian individual on various markers of inflammation, oxidative stress, and microbiome, as well as certain anthropometric and biochemical data associated with health. Although the generated data are certainly interesting, they are derived from very small cohorts, which greatly limits the perceived value of the findings. More importantly, the title and the way the data are interpreted is somewhat unscientific, grossly inflating the relevance of the results to healthiness. As such, the manuscript needs to be fully overhauled to remove these unfounded claims. Here are some specific comments:

  • The authors use a sensational title and language that are not an accurate reflection of what is in the data. This study does not show any effect on anthropometric and biochemical markers that are more obviously related to health. A suggested alternative title would be: A pilot study of the effect of L. casei from long-lived elderly on blood biochemical, oxidative, and inflammatory markers, and on gut microbiota in young volunteers.
  • The rest of the manuscript needs to be edited to remove claims that these data point to a healthier profile in response to the probiotic. In the discussion, it can be stated that some of these data point to improved immunologic, oxidative, and gut microbial profiles that have been associated with health factors. Clear citations of human studies need to be provided to support for every marker included.
  • Some bacteria are stated at the genus level (e.g., Bifidobacterium), while others are stated at the species level (e.g., E. coli) in the abstract and throughout. First of all, this is confusing and unhelpful. More importantly, referring to whole bacterial genera as beneficial or harmful in the abstract and throughout is simplistic and scientifically unfounded. Some of these genera can contain beneficial, pathogenic, or opportunistic species/strains of bacteria. A whole genus cannot be labeled as beneficial or harmful.
  • Some plots show data in scatter/bar format, while others are shown only as bars. All data should be shown in scatter/bar format.
  • Some p values are shown as exact values (e.g., P=0.023), while others are just shown as less than (e.g., P<0.05). These should be made uniform.
  • Figure 5 shows the correlation between expression of “Lactobacillus” and certain oxidative and inflammatory markers. Is this the total expression of Lactobacillus genus bacteria? If so, how does this relate specifically to the strain being examined being in this study?

Round 2

Reviewer 2 Report

The authors have responded to many of the reviewers’ comments, but some issues remain.

In many instances, the text of the manuscript states conclusions in definitive terms, whereas in reality these pertain to debatable and potentially controversial topics that need further research (hence the need for this study). For example, the opening states “Probiotic intake can effectively improve physiological health indicators.” Please change to “Probiotic intake has been shown to improve certain physiological health indicators”.

Similarly, it is stated that “This shows L. casei LTL 1879 can effectively improve immune, oxidative, and gut microbiota characteristics related to health factors.” A more scientifically sound statement would be “The results of this pilot study suggest L. casei LTL 1879 can significantly improve specific immune, oxidative, and gut microbiota characteristics related to health factors.”

The authors need to carefully go through the manuscript to remove similar statements and to only present the data without exaggerated claims, as was also suggested in the previous review.

In the abstract, please remove percentage reductions in levels of various markers. The significance level is already shown and that’s what’s important. Furthermore, the average reader may be confused by those percentages and not realize that they’re mean values.

In abstract, it is stated that “total antioxidant capacity (T-AOC) levels increased by 6.40% (p  > 0.05)”. Shouldn’t this be p<0.05 instead?

There are inconsistencies in format and style throughout the manuscript that need to be fixed. For example, in Fig. 1, the left panels show the scatter plots, but the right panels (male vs female) do not. Please make sure that all figures show the scatter plots. In Fig. 2, the font size is very different from panel to panel. Make sure that, for example, the p values are shown with the same font size. The rest of the manuscript needs to be reviewed carefully for these inconsistencies.